# Characteristics, Toxic Effects, and Analytical Methods of Microplastics in the Atmosphere

**DOI:** 10.3390/nano11102747

**Published:** 2021-10-17

**Authors:** Huirong Yang, Yinglin He, Yumeng Yan, Muhammad Junaid, Jun Wang

**Affiliations:** 1College of Marine Sciences, South China Agricultural University, Guangzhou 510642, China; hry@scau.edu.cn (H.Y.); 20202150006@stu.scau.edu.cn (Y.H.); ymyan@stu.scau.edu.cn (Y.Y.); junaid@scau.edu.cn (M.J.); 2Guangdong Laboratory for Lingnan Modern Agriculture, South China Agricultural University, Guangzhou 510642, China; 3Zhongshan Innovation Center, South China Agricultural University, Zhongshan 528400, China; 4Guangxi Key Laboratory of Marine Natural Products and Combinatorial Biosynthesis Chemistry, Biophysical and Environmental Science Research Center, Institute of Eco-Environmental Research, Guangxi Academy of Sciences, Nanning 530007, China

**Keywords:** microplastics, atmosphere, distribution, characteristics, toxicity, quantitative analysis

## Abstract

Microplastics (MPs) (including nanoplastics (NPs)) are pieces of plastic smaller than 5 mm in size. They are produced by the crushing and decomposition of large waste plastics and widely distributed in all kinds of ecological environments and even in organisms, so they have been paid much attention by the public and scientific community. Previously, several studies have reviewed the sources, occurrence, distribution, and toxicity of MPs in water and soil. By comparison, the review of atmospheric MPs is inadequate. In particular, there are still significant gaps in the quantitative analysis of MPs and the mechanisms associated with the toxic effects of inhaled MPs. Thus, this review summarizes and analyzes the distribution, source, and fate of atmospheric MPs and related influencing factors. The potential toxic effects of atmospheric MPs on animals and humans are also reviewed in depth. In addition, the common sampling and analysis methods used in existing studies are introduced. The aim of this paper is to put forward some feasible suggestions on the research direction of atmospheric MPs in the future.

## 1. Introduction

A significant number of previous studies have emphasized the ubiquitous presence of microplastics (MPs) in the oceans [1], freshwater bodies [2,3], and soil [4], along with food items, drinks [5], seasonings [6], and aquatic organisms (Figure 1). Microplastic pollution in the environment can be caused by several factors, including landfills [7]; dumping and application of sewage sludge [8]; fiber shedding of synthetic textiles; transportation (wear of tires, brakes, road signs, etc.) [9]; and other human activities, including industrial plastic pellet preproduction [10], plastic mulching and grinding in agricultural [11], fisheries, and tourism [12]. After being released into the environment through different pathways, these microplastics experience the process of degradation (physicochemical fragmentation, chemical aging, biological degradation, etc.) [11] and translocation [13] under different environmental conditions. Finally, they enter animals and humans through skin contact, oral ingestion, inhalation, and other ways and continue to get enriched [14]. After entering the body, microplastics may produce a variety of negative effects, such as decreased growth rate [15], inflammatory response [16] and oxidative stress [17], and metabolic disorders [18]. In severe cases, they can penetrate organs [19,20], tissues [21], and even cells [22], causing toxic effects. Because microplastics have a large specific surface area and strong adsorption capacity, they easily adsorb various inorganic pollutants, such as Cu, Pb, and Cd, [23] and organic pollutants, e.g., PCBs [24], PAHs [18], and polybrominated diphenyl ethers [25]. Even some microplastics themselves contain additives with a certain toxicity, and some studies have shown that microplastics combined with these pollutants pose a serious threat to organisms.

MPs in the air have been identified as particulate air pollutants and paid great attention recently [26]. However, at present, the research on environmental MPs is mainly focused on the aquatic ecosystem, and the number of studies on atmospheric MPs is limited, which is a limitation to further understanding the environmental characteristics and negative effects of atmospheric MPs. The small size of MPs, especially nanoplastics (NPs), facilitates their emissions into the air [27] and long-distance transportation [28] and can cause adverse effects on animals and humans through respiratory inhalation [29]. Critical analysis is urgently needed to open new ways of thinking about atmospheric MPs in the future.

In this review, the research progress on atmospheric MPs in recent years is summarized, including the following: (1) the global distribution of atmospheric MPs and their influencing factors, (2) the origin and fate of atmospheric MPs, (3) advances in sampling and analysis of atmospheric MPs, (4) toxicological impacts of MPs in the atmosphere on animals and humans, and last but not the least (5) the existing gaps in each part and the corresponding future research directions.

## 2. The Global Distribution of Atmospheric MPs and Associated Influencing Factors

### 2.1. Distribution Profile

At present, the relevant studies on the distribution of MPs mainly pay attention to the water and soil environment and the number of studies about atmospheric MPs is limited (Table 1). The earliest research on the distribution of atmospheric MPs can be traced back to Dris et al. collected and analyzed samples of outdoor air in Paris’s urban areas, where the concentration of MPs ranged between 29 and 280 items/m^2^/day. The size of MPs ranged from 0.1 to 5 mm, and the shapes mainly included fibers and fragments [30]. In another study, Dris et al. measured that the number of MPs in indoor air in Paris reached 5.4 items/m^3^, while that in outdoor air in the same area was only 0.9 item/m^3^, indicating indoor human activities among the major sources of MPs in indoor settings [31].

Subsequently, studies on atmospheric MPs have been carried out around the world, including a dozen countries and regions in Asia [33,34,38,39,47], Europe [36,43,45], and the Arctic [48]. The survey areas include urban sites, such as municipal areas [38], apartments [31], offices [31], industrial areas [40], terminals [35], and universities [47], as well as suburbs [32], rural areas [27], mountains [42], straits [46], estuaries [46], oceans [41,46], glaciers [28], and even the planetary boundary layer (PBL) [27]. These studies suggest that MPs exist in the atmosphere worldwide, from near the ground level up to 1.5 km high.

### 2.2. Influencing Factors

#### 2.2.1. Vertical Concentration Gradient

Similar to other air particle pollutants, the concentration of atmospheric MPs near the ground is much higher than that at high altitudes due to the influence of gravity [49]. Li et al. also demonstrated this phenomenon by collecting atmospheric plastic particle deposition on the ground and on the roof of buildings in the urban areas of Beijing, where the concentration of particles in the former was higher than that in the latter [47]. However, this does not mean that MPs in the atmosphere are not worth taking seriously. Gonzalez-Pleiter et al. found MPs in the atmospheric particulate matter samples collected by the spacecraft in the planetary boundary layer (PBL) [27]. These studies suggest that MPs are ubiquitous in atmospheric environments, ranging from near the ground to high altitudes.

#### 2.2.2. Meteorological Conditions

Dris et al. found that rainfall affects the sedimentation rate of microplastic fibers in the atmosphere of Paris. When rainfall is 0–0.2 mm/day, 2–34 items/day are recorded. When the rainfall reaches 2–5 mm/day, the amount of sediment increases to 11–355 items/day [32]. This indicates that rainfall has a significant effect on the precipitation behavior of atmospheric MPs. Boucher et al. pointed out that 7% of atmospheric MPs are transported into the ocean by wind [50], indicating that low-density atmospheric plastic particles can pollute other ecosystems through the wind as a medium. However, Prata et al. believe that atmospheric microplastic particles have properties similar to other particulate pollutants and meteorological factors such as wind, precipitation, and temperature will have an impact on their concentration changes [49]. These studies indicate that meteorological conditions are a significant factor influencing the distribution characteristics of atmospheric MPs.

#### 2.2.3. Indoor and Outdoor Atmospheric Settings

Numerous studies have shown that the concentration of MPs in the indoor air environment is much higher than that in the outdoor within the same area. Dris et al. conducted a study on the outskirts of Paris in 2015, where they analyzed microplastic fibers in the air outside and inside apartments and found that these man-made fibers are mostly polypropylene (PP). Moreover, the fiber concentration in indoor air (0.3–1.5 (0.9) items/m^3^) was significantly higher than that in outdoor air (0.4–59.4 (5.4) items/m^3^) [31]. When the outdoor environment is crowded enough, the concentration of MPs in the atmosphere can reach very high levels. Kaya et al. analyzed the concentrations of atmospheric MPs in universities and terminals with a large population in Sakarya Province, Turkey, in 2016–2017, and found concentrations of particles as high as 10,495–30,822 particles/L [35]. The high concentration of airborne MPs in indoor and crowded outdoor environments may be attributed to similar conditions of high population density and poor particle dispersion capacity.

#### 2.2.4. Regional Environmental Conditions

Varying distribution of atmospheric MPs was observed in different regions. The concentrations and types of atmospheric MPs in different urban areas have similarities and differences. Cai et al. analyzed the concentration of MPs (175–313 items/m^2^/day) in the urban air of Dongguan in 2016 and found that the content of fibers in the air was the highest (90.1%), followed by fragments (6.8%), film (2.9%), and foam (0.2%)) [33]. In the same year, Zhou et al. carried out a similar investigation in the urban area of Yantai and found that the concentration of atmospheric MPs was 130–640 particles/m^2^/day, with a higher occurrence of fiber (95.05%), followed by fragments (4.04%), film (0.73%), and foam (0.18%) [34]. The concentration of atmospheric MPs varies in different regions, which may be influenced by local meteorological conditions, the topography, the urban heat island effect, and other factors [51]. However, the types of atmospheric MPs in different regions and urban areas show high similarity. Fibers are observed as the absolute dominant shape for atmospheric MPs, while fragment, film, and foam MPs appear in significantly low quantities. It is known that microplastic fibers mainly come from synthetic textiles. Fragments may come from disposable plastic bags, film may be obtained by breaking thick plastic products, and foam may come from foamed plastics [33]. It is suggested that the atmospheric MPs in central urban areas mainly come from the shedding of synthetic textiles.

Atmospheric MP concentrations often differ between urban and suburban areas. Dris et al. collected and analyzed the deposition from the air in the urban and suburban areas of Paris during 2014–2015, and the concentration of MPs in the former (110 ± 96/m^2^/day) was much higher than that in the latter (53 ± 38/m^2^/day) [32]. The higher intensity of human activity in urban areas may have contributed to higher concentrations of atmospheric MPs. The similarity of occurrence between the two (mainly fiber) may be due to atmospheric migration that leads to regional homogenization.

However, it appears that not all suburban and remote areas have a low atmospheric microplastic distribution. Allen et al. collected atmospheric sediments from the Pyrenees Mountains in France in 2017–2018 and reported an MP concentration of 365 ± 69 items/m^2^/day. It was dominated by polystyrene and polyethylene. The occurrence was fragments (68.0%), film (20.0%), and fiber (12.0%) [42]. Ambrosini et al. collected and analyzed atmospheric sediment samples from the Forni Glacier in the Alps in 2018, and the concentration of MPs in the samples was 74.4 ± 28.3 items/kg of sediment. MP polymer types included polyester (39%), polyethylene (9%), polyamide (9%), and polypropylene (4%). Fibers accounted for 65.2%, and fragments accounted for 34.8% [52]. It can be seen that remote areas may also have high concentrations of atmospheric MPs and there are great differences in the concentrations and types among different regions.

At present, due to large differences, it is still difficult to draw a clear picture of the regional distribution of atmospheric MPs, to identify the most polluted areas, so it is necessary to master more methods to study atmospheric MPs.

### 2.3. Gaps in and Prospective Research on Distribution Characteristics of Atmospheric MPs

From the above-reviewed studies, we found that the current research on the distribution of atmospheric MPs is relatively limited and there is a lack of clear and systematic studies. To this end, we propose the following for future research:(1)It is difficult to confirm the extent of MP pollution in the atmosphere around the world. It is suggested that systematic spatial and temporal studies be conducted on the distribution of MPs in the atmosphere, to further clarify the concentrations, types, and occurrence of atmospheric MP pollution in different regions and determine the sources, distribution, and fate of atmospheric MPs in different regions.(2)We found that the experimental methods for studying atmospheric MPs in the past papers were different and no standard methods for collection and characterization of MPs were validated, which greatly reduced the experimental efficiency. In addition, the measurement criteria and units used were so varied that it is hard to intuitively make a comparison with the experimental findings of researchers using different standards (e.g., there is no way to compare the concentrations of MPs in units of m^2^/day and m^3^/day). It is suggested that the use of more efficient sampling and analysis methods be unified and the industry standards for measuring MP concentration, type, and occurrence be standardized.

## 3. Sources of Atmospheric MPs

### 3.1. Sources

#### 3.1.1. Synthetic Textiles

Synthetic textiles are a major source of atmospheric MPs. The output of synthetic textiles in the world has been growing gradually year by year. Since the annual output exceeded 60 million tons in 2016, it has been growing steadily by about 6%/year [53]. The commonly used plastic raw materials in synthetic textiles include fiber with polymers of polyamide (PA), polypropylene (PP), polyacrylonitrile (PAN), polyester (PET), polyvinyl formaldehyde (PVDF), polytetrafluoroethylene (PTFE), etc. In the use of synthetic textiles, fine fibers fall off from the fabric and are released into the air due to grinding and cutting in the textile industry, as well as wearing, washing, and drying clothes in daily life. It has been reported that thousands of fibers can be shed from a single gram of PAN fabric [54]. According to a study of De Falco et al., there is a significant correlation between the shedding of microplastic fibers (MFS) during the wearing of synthetic clothing and the type of fabric. Taking PES as an example, short silk fabrics release more MFS than filament fabrics, which may be because short fibers are easier to shed during movement, friction, and other behaviors. However, knitted garments are more likely to shed MFS than woven garments, which may be due to the looser arrangement of fabric fibers in knitted garments [55]. COVID-19 is wreaking havoc around the world, leading to a global surge in the production and use of masks and protective clothing. Mask and protective clothing materials are known to include PP, polyethylene (PE), polyurethane (PU), PTFE, PET, and ethylene side-by-side (ES) polymer plastics [56]. These anti-epidemic fabrics may become a major source of atmospheric MPs in coming years. In addition, because the inside of a mask is close to your mouth and nose, the MPs they shed can be easily inhaled.

#### 3.1.2. Transportation

Transportation also contributes a lot to atmospheric MPs, for example, in the form of wear particles from the tires and brakes of cars as well as from road surfaces and aircraft tires [9,57]. Existing research suggests that the composition of tire and road wear particles (TRWPs) is about 50% natural or synthetic polymers, which include a large number of plastic components, such as styrene–butadiene rubber (SBR). It shows that a great number of plastic particles enter the surroundings each year because of TRWP emission [58], accompanied by a certain amount of preservatives, antioxidants, desiccants, plasticizers, and other additives. According to Wagner et al., road traffic alone produces 1.327 million tons of TRWPs per year in Europe. These TRWPs can easily pollute the air environment through direct discharge or resuspension of road dust [59]. The world produces 0.2–5.5 kg of TRWPs per person per year, of which the contribution to PM10 emissions accounts for 11%. Moreover, TRWPs make up more than 50% of MP emissions in Denmark and Norway, as well as about 30% in Germany [60]. In addition, TRWPs are usually emitted in heterogeneous aggregates with other wear particles present in traffic (brake wear, road wear, etc.) [61].

#### 3.1.3. Dust

The concentration of MPs in ambient dust is very high, both in deposited dust and in suspended dust. In terms of deposited MP particles, the distribution of MPs of different shapes and sizes is very uneven under the influence of external forces (natural or human activities), and they are easily suspended in the atmospheric environment due to external forces. However, the form of MPs in floating dust is mainly fine fibers. Compared with other types of plastics, the lower-density characteristics of MPs facilitate their suspension in air [40]. Liu et al. studied and analyzed indoor and outdoor dust samples from 39 cities in China and found that PET majorly contributes to the content of MPs in indoor dust. This may be because indoor MPs are mainly derived from synthetic fabrics and PET is the main component of polyester in commonly used synthetic fabrics, which are easy to use, wash, and dry. Secondly, polycarbonate (PC), which is widely used in electronic equipment, hardware, and food packaging, can also easily fall off and enter the atmosphere [62,63].

#### 3.1.4. Other Small Sources

Atmospheric MPs are also likely to come from the degradation of large plastics, such as building materials and synthetic furniture; landfills; synthetic particles from gardening; and industrial and other emissions-related activities. However, compared with the major sources of MPs such as synthetic textiles, transportation, and dust, the actual contribution of these sources to atmospheric MPs may be very small and remains in the stage of idea and speculation, without the availability of data [32,33,38].

#### 3.1.5. Gaps in and Prospective Research on the Sources of Atmospheric MPs

From the above series of articles, it is found that the current articles on the sources of atmospheric MPs are mostly focused on synthetic textiles, transportation, dust deposition, and resuscitation, while other sources are often ignored, and the contribution of these sources to atmospheric MPs cannot be quantified. Detailed data on their pollution concentrations, types, and occurrence are scarce. In the analysis of the sources of atmospheric MPs in many pieces of literature, the proportion of unknown sources in the experimental results is relatively high, indicating that there is still a considerable part of atmospheric MP sources that is unclear. In addition, many research methods to identify the origin remain in the characterization and chemical composition analysis. To this end, we propose the following for future research:(1)Continue to optimize the methods and tools for atmospheric MP characterization and component analysis and identification to more clearly identify the sources of MPs and avoid unclear and inaccurate source identification caused by rough differentiation.(2)Establish a pollution source localization method suitable for atmospheric MPs, which can trace the source more accurately than characterization or component analysis.

### 3.2. Transportation and Fate of Atmospheric MPs

#### 3.2.1. Migration

Under certain conditions, MPs in the air can migrate to other ecological environments (soil, water, etc.). Their behavior, transportation, concentration, and deposition are affected by various factors, such as vertical pollution concentration gradient (VPCG) [64,65], meteorological conditions (rainfall, precipitation, temperature, humidity, wind (occurrence, velocity, duration, intensity, and direction), etc.) [39,42,43,66], population density, human activities [32,39], urban topography, thermal cycling [51,64], local elevation, and geographical environment [39]. Some researchers have found that MPs travel long distances in the air. For example, MPs are found in extremely remote areas and even in the snow and ice of high-altitude glaciers. It is speculated that MPs cause cross-border and global pollution through air and wind currents [27,28].

#### 3.2.2. Inhalation

Since the MP concentration indoors is much higher than that outdoors, atmospheric MPs enter the body through human inhalation [31]. Indoor air is an essential source of human exposure to airborne MPs because people stay indoors for longer periods and dispersing machines are less capable of removing plastic particles [49]. Factors affecting MP behavior and transmission in indoor air are ventilation, airflow, and room spacing [67]. The concentration of MP particles indoors is higher than that outdoors, which may be influenced by textiles, furniture, building materials, and human activities [31]. Compared with MPs in other environments, MPs indoors are more easily inhaled directly and continuously and cause health risks [49,53].

One of the earliest criteria for determining airborne MPs is 0.3–1.5 particle/m^3^ outdoors and 0.4–56.5 particles per cubic meter indoors (33% polymer) [31]. According to statistics, each person inhales between 26 and 130 MP particles from the air per day [49]. Based upon air samples taken from mannequins, men who exercise lightly can expect to inhale 272 particles/day [44]. Estimates vary depending on sampling methods and space use factors.

#### 3.2.3. Gaps in and Prospective Research on the Destinations of Atmospheric MPs

Based on the above discussion, the studies on the fate of MPs in the atmosphere are limited. So far, only limited studies have traced the atmospheric migration paths of MPs; therefore, it is nearly impossible to quantify the various environmental factors and human activities affecting the behavior and transmission of atmospheric MPs. Further, the quantification and characterization of atmospheric MPs in different parts of the human body and associated health impacts is challenging. To this end, we propose the following for future research:(1)Further explore the factors affecting the fate of atmospheric MPs and understand the different destinations of MPs in the atmospheric environment under different conditions.(2)Establish a spatial model and related software suitable for integrating the diffusion and migration trajectories of atmospheric MPs and the pollutants adsorbed by them.(3)Investigate the difference in the quantities and proportions of MPs absorbed by people in different areas and under different conditions in the same area as well as body burden and associated risks of atmospheric MPs.

## 4. Toxic Effects

After the body inhales MPs, they enter the lungs along the trachea, enter the blood vessels through migration, and then spread through the circulatory system throughout the body, causing various degrees of toxic effects on the cells, tissues, organs, and systems of the body (Figure 2) [19,68,69,70]. 

### 4.1. Inhalational-Based Toxicity

MPs and NPs in the air mainly enter animals and humans through inhalation and first contact with the respiratory system. Due to the clearance mechanisms of the respiratory barriers (such as the nasal cavity, trachea, the bronchus, and alveolar macrophages), a considerable proportion of these plastic particles cannot enter the body and form deposition, and the deposition coefficient is mainly affected by the particle size, density, etc. [49]. Smaller nanoparticles (<2.5 μm) can avoid the clearance mechanism and penetrate the lung and respiratory barrier [19,68,69,70]. Macrophages can deposit plastic particles in the respiratory system, remove these from the respiratory system, or help them migrate in the respiratory system and allow them to enter the circulatory system, leading to dust metastasis. Further, the presence of MPs and NPs in the respiratory system and their large surface area can also lead to the release of chemokines that affect the migration of macrophages, resulting in particle overload [68]. The influx and inflammation of neutrophils associated with polystyrene granules (64 nm) in rat lungs, as well as the expression of pro-inflammatory genes in epithelial cells, are caused by granule-induced oxidation [71]. Due to the large specific surface area of MPs, a large number of oxidizing substances (such as metals) will be adsorbed on the surface of MPs, which may produce excessive reactive oxygen species and lead to excessive antioxidant reaction in vivo, namely oxidative stress [72,73].

There are only a few pieces of research on the toxic effects of atmospheric MPs on animals and humans. Inhalation of MPs has the most significant toxic effects on the respiratory system. Studies have shown that PVC prepared by emulsion polymerization (2 mm) has significant cytotoxic and hemolytic effects on rat and human lung cells in vitro [74]. Xu et al. 2002 [74] evaluated the toxic effects of PS-NPs on human alveolar epithelial cells and found that PS-NPs can rapidly internalize into cells; significantly reduce cell viability; affect cell cycle and apoptosis, as well as related gene transcription and protein expression; and promote inflammatory response. The smaller the diameter, the faster the corresponding velocity. Lim et al., using metabonomics to investigate the toxic effects of PS-NPs on bronchial epithelial cells, found that nanoparticles interfere with cell energy metabolism, accompanied by oxidative stress, and mediate the increase of intermediate metabolites to reduce cell resistance to toxicity [75]. Dong et al. found that PS-MPs can induce cytotoxicity and inflammation in human lung epithelium by inducing reactive oxygen species formation. Low concentrations of PS-MPs disrupt the lung barrier, while high concentrations of PS-MPs also induce decreased levels of α 1-antitrypsin, which increases the risk of chronic obstructive pulmonary disease [76]. Paget et al. explored particle internalization and cell damage after aminated PS-NPs acted on human lung epithelial cells and macrophages. Cells in the experimental group showed high glutathione depletion, excessive reactive oxygen species, and significant DNA damage. It is suggested that micro- and nanoplastics may exhibit strong genotoxicity when absorbed and internalized by the respiratory system [77]. In in vitro experiments, seven functional groups with different charges were added to an aminated PS-NPs and eight PS-NPs, including the original particles, were sent into different rats by pharyngeal aspiration. Most of the particles were phagocytized by alveolar macrophages and showed different acute lung inflammation. The particle potential showed excellent correlation with pneumonia-related parameters, indicating that surface charge is a key factor affecting lung inflammation induced by micro- and nanoplastics [78]. 

Respiratory lesions have been found in workers exposed to synthetic textiles, fusing, vinyl chloride, or PVC, which is quite different from the general population [79,80,81,82,83]. Inhaled MPs and NPs are not easily cleared by the human lungs, and they may stay in the lungs for a considerable period, which can cause an inflammatory response in the lungs [53]. It is reported that artificial glass fiber can cause DNA damage, which can induce cancer [84]. In short, Inhaling MPs can have toxic effects on animals and humans.

### 4.2. Other Toxic Effects

The inhaled MPs can also have negative effects on other systems of the body, such as diffusion or translocation. In vivo experiments have confirmed that microplastic particles can enter the circulatory system through the migration of macrophages after inhalation [68]. MPs in the circulation system may cause inflammation, vascular occlusion [85], and other blood toxicity [86]. In vitro experiments showed that polystyrene nanoparticles cause red blood cell (RBC) aggregation, while polypropylene particles increase hemolysis [87]. Inhaled MPs may enter the gastrointestinal tract by clearing upper respiratory tract cilia [53]. MPs entering the digestive system may alter the permeability of intestinal epithelial cells and cause changes in microbial composition [88]. Recent reports have found that after maternal lung exposure to NPs, the plastics can enter the placenta and the fetus through translocation and get deposited in the fetal liver, kidney, nervous system, and circulatory system. [21]. There is little research on the toxic effects of atmospheric MPs on systems other than the respiratory system, and more research must be done to explore their effects on animals and humans.

### 4.3. Joint Toxic Effects

MPs usually contain various additives, e.g., catalysts (organotin), flame retardants, polybrominated diphenyl ethers [89], antioxidants (nonylphenol), antibacterial agents (triclosan), and plasticizers (phthalate PAEs) [90]). All of them are harmful to animals and humans. In addition, MPs adsorb many inorganic pollutants in the environment, e.g., Au [16] and Cu [91], and organic pollutants, e.g., polycyclic aromatic hydrocarbons [18] and polychlorinated biphenyls [92]. These pollutants are mutagenic and carcinogenic substances that are widely present in the environment. In addition, many plastic monomers themselves (polystyrene (PS), polyvinyl chloride (PVC), etc.) have mutagenic and carcinogenic toxic effects on animals and humans [93]. Exposure of animals and humans to additives or adsorbents in these plastic particles results in combined toxicity (Table 2). MPs in the atmosphere are most likely to combine with other pollutants in atmospheric environments (such as POPs and Cu) and serve as carriers for the long-distance transport of these pollutants in the atmosphere [53]. Some articles have studied the negative effects of MPs combined with these pollutants on animals and the human body, such as a series of oxidative stress [16], inflammatory reaction [91], and metabolic disorders [18]. There is a risk of mutation and carcinogenesis [94]. Most of these studies have involved oral ingestion or in vitro studies. However, there are few studies on the toxicity mechanism of these binding substances after they enter animals and humans through inhalation, and more experimental data are needed.

### 4.4. Gaps in and Prospective Research on the Toxic Effects of Atmospheric MPs on Animals and Humans

In conclusion, studies on the quantitative analysis and toxicity mechanism of atmospheric MPs inhaled by animals or humans are scarce and these aspects need to be further studied. To this end, we propose the following for future research:(1)In vivo experiments should be conducted to explore the different negative effects of MPs or NPs with different physical or chemical properties (such as different types, sizes, occurrences, crystallinity, and surface charge) on animal and human health after inhalation. An animal model should be established to study the movement trajectory and deposition proportion of and harmful substances released by atmospheric MPs in the body.(2)To better understand the harmful additive impact of atmospheric MPs adsorbed with other pollutants as a pollutant internalization carrier, more research into the toxic additive effect of the two is required.

## 5. Existing Analytical Methods and Gaps in Measuring Atmospheric MPs

There are two main sampling methods for atmospheric MPs commonly used. One way is to collect the passive fallout from the atmosphere and filter it. The depositions in the atmosphere are collected through a non-plastic funnel (such as stainless steel or glass) the pipe of which drops into a glass collection bottle below. MPs can then be easily filtered out of the sediment [32,33,42,43]. Another method is active pump sampling and filtration, mainly through a set of pumping and filtration system, in which air is collected through the pump unit and then filtered through the filter to retain the plastic particles [31,38,40]. After the sample is collected, different efficient quantitative analysis methods can be used to analyze the types and sizes of particles (Table 3).

Although a set of primary collection and analysis methods has been established, there are still many limitations and deficiencies. The above methods can only identify some common types of plastic particles at the ground level or near the ground level [27]. Moreover, it is difficult to accurately identify the size of the nanoplastics or the types of particles adsorbed by organic matter on their surfaces [101,103,104]. To this end, we propose the following for future research:(1)To further develop and research some efficient methods and instruments. On the one hand, a great number of MPs should be sampled and accurately identified in a short period. On the other hand, we should be able to further identify more types of plastics and plastics of smaller sizes through microfiltration and various pollutants adsorbed on them.(2)To develop a set of uniform standard methods for sampling and identification. For this, scientific data generated in different regions for atmospheric MPs should be compared.

## 6. Conclusions

MPs have been well studied in marine and freshwater environments, but MPs in the atmosphere have received little attention from researchers and society. Microplastics in the atmosphere enter the body mainly through inhalation and further systemic exposure, causing toxic reactions and disorders in various organs and systems and even posing a potential risk of cancer to animals and humans.

Due to the lack of practical methods for detection and analysis, we are still unable to gain a more detailed understanding of the global distribution, sources, and fate of atmospheric MPs, let alone further elucidate the mechanisms of toxic action of atmospheric MPs on animals and humans. We suggest that the scientific community conduct in-depth research on atmospheric MPs in the future, especially to explore relevant sampling and detection methods and establish a common industry standard. Further quantitative analysis of atmospheric MPs of different types and properties will be conducted to explore the toxicity mechanism and additive effect of their combination with other pollutants through in vivo experiments.

## Figures and Tables

**Figure 1 nanomaterials-11-02747-f001:**
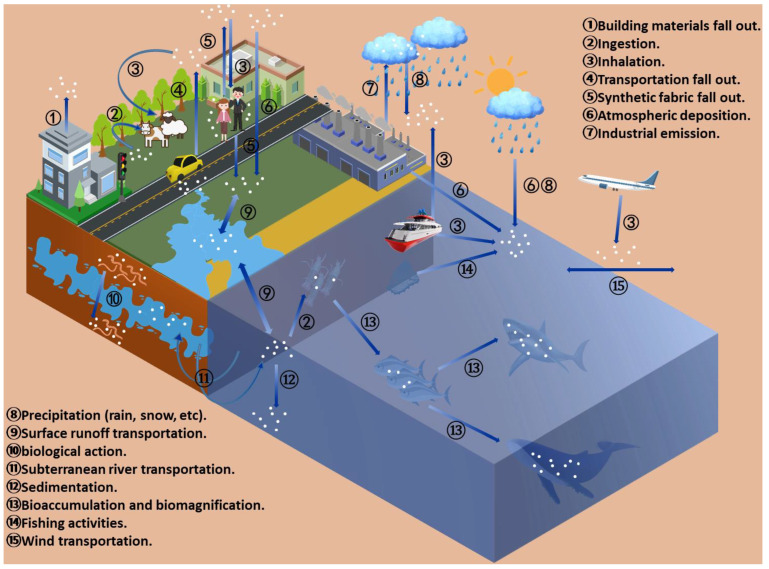
Cycle of microplastic pollution in ecosystems.

**Figure 2 nanomaterials-11-02747-f002:**
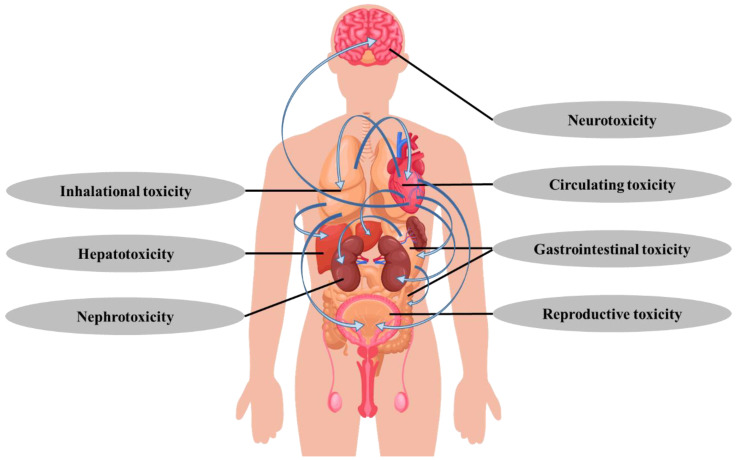
Toxic effects of atmospheric microplastics on different organs and systems.

**Table 1 nanomaterials-11-02747-t001:** Distribution and abundance of plastics in the atmosphere.

Location	Year	Sample Type	MP Type	Shape	Concentration (Item/Particle Number)	Size	Reference
Paris	2014	Urban outdoor air deposition	NA	Fiber, fragment	29–280/m^2^/day	0.1–5 mm	[30]
Paris	2014–2015	Urban outdoor air deposition	NA	Fiber	110 ± 96/m^2^/day	0.05–5 mm	[32]
Paris	2014–2015	Suburban outdoor air deposition	NA	Fiber	53 ± 38/m^2^/day	0.05–5 mm	[32]
Paris	2015	Urban indoor air	PA, PP, PE	Fiber	0.4–59.4 (5.4)/m^3^	0.05–3.25 mm	[31]
Paris	2015	Urban outdoor air	PA, PP, PE	Fiber	0.3–1.5 (0.9)/m^3^	0.05–1.65 mm	[31]
Dongguan	2016	Urban outdoor air deposition	PE, PP, PS	Fiber, foam, fragment, film	175–313/m^2^/day	Minimum: <0.2 mmMaximum: >4.2 mm	[33]
Yantai	2016	Urban outdoor air deposition	PET, PVC, PE, PS	Fiber, fragment, film, foam	2.33 × 10^13^/160 km^2^/year	0.05–1 mm	[34]
Sakarya	2016–2017	Crowded area outdoor air	PA, PUR, PE, PP, PES	Fiber, fragment	9067–30,793/L	0.05–0.5 mm	[35]
Edinburg	2017	Indoor air of houses	NA	Fiber	5 ± 33/sample	NA	[36]
Trent catchment	2017–2018	River catchment air deposition	NA	Fiber	2.9–128.42/m^2^/day	NA	[37]
Shanghai	2018	Municipal outdoor air	PET, PE, PES, PAN, PAA, RY, EVA, EP, ALK	Fiber, fragment, granule	0–4.18 (1.42 ± 1.42)/m^3^	23.07–9554.88 μm	[38]
Shanghai	2019	Urban outdoor air	PET, EP, PE, ALK, RY, PP, PA, PS	Fiber, fragment, microbead	0–2 (0.41)/m^3^	12.35–2191.32 μm	[39]
Asaluyeh	2017	Urban and industrial outdoor air	NA	Fiber, fragment, film	0.3–1.1/m^3^	2–100 μm	[40]
West Pacific Ocean	2018–2019	Ocean air	PET, PE, PE-PP, PES, ALK, EP, PA, PAN, PR, PMA, PP, PS, PVA, PVC	Fiber, fragment, granule, microbead	0–1.37 (0.06 ± 0.16)/m^3^	16.14–2086.69 μm	[41]
Pyrenees	2017–2018	Remote air deposition	PS, PE, PP, PVC, PET	Fiber, fragment, film	365 ± 69/m^2^/day	Minimum: <0.025 mmMaximum: >2.6 mm	[42]
Hamburg	2017–2018	Urban and rural outdoor air deposition	PE, EVA, PTFE, PVA, PET	Fragment, fiber	136.5–512/m^2^/day	Minimum: <0.063 mmMaximum: >0.3 mm	[43]
Aarhus	2017	Indoor air of apartments	PES, PA, PS, PE, PUR	Fragment, fiber	1.7–16.2 (9.3 ± 5.8)/m^3^	4–398 μm	[44]
London	2018	Urban outdoor air deposition	PAN, PES, PA, PP, PVC, PE, PET, PS, PUR, petroleum, resin, acrylic	Fragment, film, granule, foam	771 ± 167/m^2^/day	75–1080 μm	[45]
Karimata Strait	2019	Strait air	PET	Fiber	0–0.8/100 m^3^	382.15	[46]
Pearl River Estuary	2019	River estuary air	PA, PEP, PET, PP	Fiber	3–7.7/100 m^3^	288.2–1117.62 μm	[46]
South China Sea	2019	Ocean air	PET, PEVA, PP	Fiber, fragment	0–3.1/100 m^3^	286.1–1861.78 μm	[46]
East Indian Ocean	2019	Ocean air	PAN-AA, PET, PP, PR	Fiber, fragment	0–0.8/100 m^3^	58.591–988.37 μm	[46]
Beijing	NA	Urban outdoor air deposition	NA	Fiber	Surface layer: 5.7 × 10^−3^/mLRoof: 5.6 × 10^−3^/mL	5–200 μm	[47]
Alcalá de Henares- Guadalajara, Valladolid	2020	Rural and sub-rural PBL air	PET, PA, acrylic	Fiber	Rural area: 1/sampleSub-rural area: 3/sample	0–9.8 μm	[27]
Guadalajara	2020	Urban PBL air	PU, PS, PA, acrylic	Fragment, fiber	6/sample	NA	[27]
Madrid	2020	Urban PBL air	PA, PU, PET, PB, PE, PP	Fragment, fiber	12/sample	NA	[27]

PA: polyamide; PP: polypropylene; PE: polyethylene; PS: polystyrene; PET: polyethylene terephthalate; PVC: polyvinyl chloride; PUR: polyurethane; PES: polyester; PAN: polyacrylonitrile; PAA: poly(N-methyl acrylamide); RY: rayon; EVA: ethylene vinyl acetate; EP: epoxy resin; ALK: alkyd resin; PR: phenoxy resin; PTFE: Teflon; PVA: polyvinyl acetate; PEVA: poly(ethylene-co-vinyl acetate); PEP: poly(ethylene-co-propylene); PAN-AA: poly(acrylonitrile-coacrylic acid); PB: polybutadiene; PU: polyurethane; PBL: planetary boundary layer; NA: not available.

**Table 2 nanomaterials-11-02747-t002:** Toxic effects of chemicals in microplastics or nanoplastics on animals.

Classification	Chemicals	Affected Species	Resulting Toxicity	Reference
Ingredient	C_6_H_6_	*Human*	Mutagenic risk	[94]
	C_6_H_5_OH	*Human*	Mutagenic risk	[94]
	BD	*Human*	Cancer risk	[94]
	VCM	*Human*	Cancer risk	[94]
Adsorption	Au	*Danio rerio*	Embryo: ① Oxidative stress ② Inflammation	[16]
	CBz	*Mytilus galloprovincialis*	Larva: Excessive oxidation of digestive glands	[95]
	Cu	*Danio rerio*	Inflammation	[91]
	PAHs	*Danio rerio*	Metabolic disorders	[18]
	PCBs	*Human*	Neurotoxicity	[92]
Dyestuff	Pyrene	*Mytilus galloprovincialis*	① Immune responses ② Lysosomal compartment dysfunction ③ Peroxisome dysfunction ④ Antioxidant system disruption ⑤ Neurotoxic effects	[96]
Flame retardants	PBDEs	*Human*	① Thyroid homeostasis disruption ② Neurotoxicity ③ Reproductive changes ④ Cancer risk	[89]
Paint coat	TiO_2_	*Caenorhabditis elegan*	Oxidative stress	[97]
Plasticizer	BPA	*Danio rerio*	Neurotoxicity	[98]
		*Rat*	Estrogen disorder	[99]
		*Human*	① Enzyme abnormality and damage of the liver ② Pancreatic cell dysfunction ③ Thyroid hormone disorder ④ Promotion of obesity ⑤ Cardiovascular disease ⑥ Low insulin levels	[99]
	DEHP, MEHP	*Rat*	Inhibition of estrogen levels	[90]
	PAEs	*Human*	① Increased risk of cardiovascular disease ② Reproductive system disruption	[90]

BD: butadiene; VCM: vinyl chloride monomer; CBz: carbamazepine; PAHs: polycyclic aromatic hydrocarbons; PCBs: polychlorinated biphenyls; PBDEs: polybrominated diphenyl ethers; BPA: bisphenol A; DEHP: dioctyl phthalate; MEHP: mono(2-ethylhexyl) phthalate; PAEs: phthalic acid esters.

**Table 3 nanomaterials-11-02747-t003:** Outstanding methods for the analysis of microplastics and nanoplastics in the atmospheric environment.

Methods	Tools	Medium	Plastic Components	Optimal Size	Advantages	Disadvantages	References
Spectral analysis	FT-IR	Water, oil, air	RY, PE, PET, PAA	>20 μm	① It does not destroy the sample. ② Pretreatment is simple. ③ The type of plastic particles can be determined.	It is difficult to identify the types of plastic particles that are aged or have contaminated surfaces.	[100,101,102,103]
	RM	Water, air	PA, PC, PE, PP, PS, PET, PVC, PMP, PCL, PMMA	0.5–20 μm	① It does not destroy the sample. ② It supports nano-sample imaging. ③ It supports low sample amount identification. ④ It is environmentally friendly.	① The measurement time is long. ② Fluorescence interference is easy to produce. ③ The signal-to-noise ratio is low. ④ The use of laser as the light source leads to background emission and sample degradation.	[104,105,106,107,108]
Thermal analysis	TGA-DSC	NA	PE, PP, etc.	NA	① The operation is simple. ② Less sample is required (1–20 mg). ③ Accuracy is high.	① It is difficult to distinguish the polymers with similar transition temperatures. ② It is difficult to identify copolymers. ③ The samples are destroyed. ④ It cannot identify the morphology, size, and quantity of the plastic particles.	[108,109,110,111,112]
	Py-GC-MS	NA	PA, PC, PE, PS, PP, rubber, PET, PVC, PMMA	NA	① Less sample is required (5–200 μg). ② The microplastic type and weight and additives can be identified simultaneously without pretreatment. ③ The accuracy is high. ④ It recognizes copolymers.	① The samples are destroyed. ② It cannot identify the morphology, size, and quantity of the plastic particles.	[110,111,112,113,114,115]
	TED-GC-MS	NA	PA, PE, PP, PS, PET	NA	It involves simple pretreatment and operation.		[108,110,112,116]
Other analytical methods	SEM-EDS	Majority	Majority	≥1 nm	① Imaging is at the nanoscale. ② Elements can be identified	① It is expensive. ② Work efficiency is low.	[108,110,112,117,118]
	MS	Majority	Majority	≥1 nm	It can identify the structure, molecular weight, degree of polymerization, functional group, and end group structure of the plastic particles.	Different samples require different ionizing reagents (poor applicability).	[108,119,120]
	XPS	Majority	Majority	>10 nm	It can identify elemental composition and content, chemical state, molecular structure, and chemical bonds.	It cannot identify the nanoplastic types definitely.	[121,122]
	RMR	Water, oil	Majority	>50 μm	① The cost is low. ② It is convenient for real-time field detection.	① It is only used to detect the concentration. ② It requires specific calibration samples.	[123]

FT-IR: Fourier-transform infrared spectroscopy; RM: Raman spectroscopy; TGA-DSC: thermogravimetric analysis-differential scanning calorimeter; Py-GC-MS: pyrolysis gas chromatography-mass spectrum; TED-GC-MS: thermal desorption gas chromatography-mass spectrum; SEM-EDS: scanning electron microscope-energy dispersive spectrometer; MS: mass spectrometry; XPS: X-ray photoelectron spectroscopy; RMR: resonance microwave reflectometry. RY: rayon; PE: polyethylene; PET: polyethylene terephthalate; PAA: polyacrylic acid; PA: polyamide; PC: polycarbonate; PP: polypropylene; PS: polystyrene; PVC: polyvinyl chloride; PMP: polymethylpentene; PCL: polycaprolactone; PMMA: polymethylmethacrylate; NA: not available.

## Data Availability

Not Applicable.

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
