# Peer review of "Characteristics, Toxic Effects, and Analytical Methods of Microplastics in the Atmosphere"

_nanomaterials, 2021, doi:10.3390/nano11102747_

Round 1

Reviewer 1 Report

Congratulations to the authors in preparing this important review. The manuscript is well organised and most relevant references are included. The different areas of the review are included in separate and appropriate sections. The tables included are an effective way of presenting large data and were important to include in the manuscript.

However, had the sections on toxicity been expanded,It would have made the review even more substantial.  

Author Response

Reviewer 1:

Q: Congratulations to the authors in preparing this important review. The manuscript is well organised and most relevant references are included. The different areas of the review are included in separate and appropriate sections. The tables included are an effective way of presenting large data and were important to include in the manuscript.

However, had the sections on toxicity been expanded, It would have made the review even more substantial.  

 A: Thank you for your suggestion. Based on your suggestion to expand the toxic part of the manuscript, we reviewed the toxicity studies of atmospheric microplastics in recent years and enriched the manuscript, hoping to make you think there are brand new feeling.

Reviewer 2 Report

Although the paper is interesting, bringing to the readers' attention the research progress made in recent years on atmospheric microplastics in terms of characteristics, toxic effects, and analytical methods of detection in the atmosphere, my opinion is that this paper does not fall within the stated purpose of this journal, namely the preparation, characterization, and applications of NANOMATERIALS. I think a journal covering materials in general, for example, Materials (MDPI) would be much more appropriate.

Author Response

Reviewer 2:

Q: Although the paper is interesting, bringing to the readers' attention the research progress made in recent years on atmospheric microplastics in terms of characteristics, toxic effects, and analytical methods of detection in the atmosphere, my opinion is that this paper does not fall within the stated purpose of this journal, namely the preparation, characterization, and applications of NANOMATERIALS. I think a journal covering materials in general, for example, Materials (MDPI) would be much more appropriate.

A: Thank you for your suggestions. The special issue "Nanoparticles in the Environment and Nanotoxicology" aims to gather recent novel research findings of various types of nanoparticles on the determination, detection, and degradation in the environment as well as the toxicity and risk assessment of nanoparticles. The theme of the manuscript is the characteristics, toxicity and analysis of atmospheric micro and nano plastics, among which environmental characteristics and toxicology are the main content. We believe that the content of the manuscript should be consistent with the requirements of special issue.

Reviewer 3 Report

Dear authors,

Thank you very much for the interesting review "Characteristics, toxic effects and analytical methods of microplastics in the atmosphere".

Your review represents a significant contribution to the field of environmental pollution. The paper is very well organized with significant numbers of references, data, and figures.

The only difficulty for reviewing your paper was the fact that the paper is not arranged according to instructions for authors, there are no lines numbers in the text.

Although I used my time to carefully read your paper and I can say well done.

You have minor language corrections in the text and the Abstract and Conclusion part needs to be rewritten to gain more scientific sound with the most important data, instead of storytelling.

After these modifications, I will recommend acceptance of the paper.

All the best and stay safe

Author Response

Reviewer 3:

Q:

Dear authors,

Thank you very much for the interesting review "Characteristics, toxic effects and analytical methods of microplastics in the atmosphere".

Your review represents a significant contribution to the field of environmental pollution. The paper is very well organized with significant numbers of references, data, and figures.

The only difficulty for reviewing your paper was the fact that the paper is not arranged according to instructions for authors, there are no lines numbers in the text.

Although I used my time to carefully read your paper and I can say well done.

You have minor language corrections in the text and the Abstract and Conclusion part needs to be rewritten to gain more scientific sound with the most important data, instead of storytelling.

After these modifications, I will recommend acceptance of the paper.

All the best and stay safe

 A: Thank you for your suggestions. As for the missing lines numbers, it is our mistake. We have made a supplement in the latest manuscript for you to read again. We have carefully read your suggestions on the abstract and conclusion part of the manuscript and modified them, hoping to make you feel different.

Round 2

Reviewer 2 Report

Line 323-326: The reference is missing after the phrase “Xu et al.  evaluated the toxic effects of PS-NPs on human…….”

Author Response

The reference in Line 323-326 has been added.